# #WhatIEatinaDay: The Quality, Accuracy, and Engagement of Nutrition Content on TikTok

**DOI:** 10.3390/nu17050781

**Published:** 2025-02-24

**Authors:** Michelle Zeng, Jacqueline Grgurevic, Rayan Diyab, Rajshri Roy

**Affiliations:** 1Nutrition and Dietetics, Susan Wakil School of Nursing and Midwifery, Faculty of Medicine and Health, The University of Sydney, Sydney, NSW 2050, Australia; michelle.zeng@sydney.edu.au (M.Z.); jgrg5772@uni.sydney.edu.au (J.G.); rdiy2498@uni.sydney.edu.au (R.D.); 2Charles Perkins Centre, The University of Sydney, Sydney, NSW 2050, Australia

**Keywords:** nutrition, misinformation, social media, engagement metrics, adolescents, young adults

## Abstract

**Background:** Social media platforms such as TikTok are significant sources of nutrition information for adolescents and young adults, who are vulnerable to unregulated, algorithm-driven content. This often spreads nutrition misinformation, impacting adolescent and young adult health and dietary behaviors. **Objectives:** While previous research has explored misinformation on other platforms, TikTok remains underexamined, so this study aimed at evaluating the landscape of nutrition-related content on TikTok. **Methods:** This study evaluated TikTok nutrition-related content by (1) identifying common nutrition topics and content creator types; (2) assessing the quality and accuracy of content using evidence-based frameworks, and (3) analyzing engagement metrics such as likes, comments, and shares. **Results:** The most common creators were health and wellness influencers (32%) and fitness creators (18%). Recipes (31%) and weight loss (34%) dominated the list of topics. When evaluating TikTok posts for quality, 82% of applicable posts lacked transparent advertising, 77% failed to disclose conflicts of interest, 63% promoted stereotypical attitudes, 55% did not provide evidence-based information, 75% lacked balanced and accurate content, and 90% failed to point out the risk and benefits of the advice presented. A total of 36% of posts were considered completely accurate, while 24% were mostly inaccurate, and 18% completely inaccurate. No statistical significance was associated between the level of accuracy or evidence and engagement metrics (*p* > 0.05). **Conclusions**: TikTok prioritizes engagement over accuracy, exposing adolescents to harmful nutrition misinformation. Stricter moderation and evidence-based nutrition content are essential to protect adolescent and young adult health. Future research should explore interventions to reduce the impact of misinformation on adolescent dietary behaviors and mental well-being.

## 1. Introduction

Social media platforms, particularly TikTok, have become significant sources of health and nutrition information for young users. TikTok’s popularity among adolescents and young adults makes it a powerful tool for disseminating health information, especially on topics related to diet, nutrition, and weight management [1,2,3,4]. TikTok has over 1 billion monthly active users, with approximately 63% of its user base being between 10 and 29 years old, making it a highly influential platform for adolescents [5]. Unlike Instagram and YouTube, which emphasize curated content and long-form videos, TikTok’s algorithm prioritizes short, engaging videos based on virality rather than content credibility [3]. This increases the potential for misinformation to spread rapidly, particularly in health-related topics. “What I Eat in a Day” type videos are prevalent on this platform and often spread misinformation about diet and health and influence audience attitudes and behaviors with weight-normative messaging [6]. However, the quality and accuracy of content vary widely, with a significant portion of health and nutrition information not provided by experts. This has led to the spread of weight-normative and potentially harmful messages, contributing to disordered eating behaviors and body dissatisfaction among young users [1,7,8].

While some health professionals use TikTok to share credible diet-related content, their presence is relatively limited compared with the abundance of non-expert opinions [4,9,10]. Health professionals’ videos tend to be more credible and preferred by users, yet much of the platform is saturated with non-expert voices. TikTok is particularly influential among young women and adolescents of color, who often rely on the platform for health information. These demographics face heightened risks from misinformation, which can influence their health-related behaviors and perceptions [2,4]. TikTok usage is also influenced by socioeconomic factors, with lower-income populations relying on social media for health information due to barriers in accessing professional healthcare [11,12]. Additionally, users frequently employ “algospeak” to bypass content moderation algorithms, further complicating the platform’s ability to ensure accurate and context-sensitive information. This practice disproportionately affects marginalized communities by spreading biased or inaccurate messages [13].

The risks of misinformation on TikTok include the adoption of harmful dietary practices, body image issues, and mental health concerns. Exposure to unhealthy food marketing on platforms such as Instagram and YouTube has already been shown to influence adolescents’ food preferences, contributing to poor dietary habits and obesity risks [14,15,16]. Similarly, TikTok exposes adolescents to extreme dieting content and harmful body image ideals. The platform often glorifies weight loss, promotes diet culture, and provides a narrow view of health, primarily driven by non-expert creators [1]. Videos from content creators claiming expertise without credentials further amplify misinformation, while humorous and viral content often overshadows accurate information [17,18,19,20].

Despite its significant influence, there is a notable lack of research regarding nutrition-related content on TikTok and how misinformation operates on the platform. This gap likely stems from TikTok being a relatively new platform compared with Facebook and Instagram, as well as the rapidly evolving nature of social media [1,3,9,21]. TikTok’s unregulated and algorithm-driven promotion of content creates significant risks for the spread of nutrition misinformation, particularly amongst its adolescent and young adult audience [1,3,16,21]. TikTok has recently updated its policies on beauty filters and content transparency, prompting legal discussions in countries, such as Australia, over the potential impact of digital manipulation on adolescent well-being [22]. However, the platform also offers an opportunity for health professionals to leverage TikTok’s popularity to share reliable information to positively influence dietary behaviors [9].

Therefore, this study aimed at evaluating the landscape of nutrition-related content on TikTok, identifying the common topics and content creators, assessing the quality, accuracy, and evidence of content evidence-based frameworks and analyzing the engagement metrics of nutrition-related posts. By evaluating these aspects, this study sought to better understand the current state of nutrition-related content and misinformation on TikTok in order to better inform the development of strategies to mitigate the harmful spread of nutrition misinformation and to promote credible, evidence-based information.

## 2. Materials and Methods

### 2.1. Study Design

A cross-sectional study was used to investigate the prevalence, characteristics, and engagement metrics of nutrition-related content on TikTok. Data collection took place in April 2024 with nutrition-related posts published between September 2023 and March 2024 being identified and analyzed.

### 2.2. Sample Selection and Randomization

The PRISMA-ScR protocol was applied to the study to identify TikTok posts relevant to this study (Figure 1) [23]. A search strategy was piloted to identify relevant posts using popular nutrition-related hashtags, including #healthyeating, #healthyfood, #diet, and #weightloss. Based on the results of the pilot, adjustments were made to the hashtags used in order to ensure that the search strategy delivered more nutrition-relevant content. For instance, #weightloss was adjusted to #weightlosstips (see Appendix A for the full search strategy). Posts were viewed using newly created accounts in a private browser in order to minimize the influence of TikTok’s algorithm, which can adapt and tailor content to target users’ prior activity and therefore bias the results.

### 2.3. Data Collection

After identifying 1054 eligible posts following the screening process, we applied a random sampling approach to select a final sample of 250 posts for detailed analysis. Data were collected for the 250 sample posts using Microsoft Excel [24] for details, including “username”, “URL” and “upload date”. Engagement metrics included the number of “likes”, “comments”, “shares”, and “saves” alongside the subscription numbers for the content creators associated with the sample posts. After collecting TikTok posts through keyword searches, the posts were screened based on predefined inclusion and exclusion criteria, including relevance to nutrition, accessibility, and language. Microsoft Excel’s functions were used to remove duplicates, assign unique identifiers, and randomly select posts for detailed analysis, ensuring an unbiased sample. Excel was primarily used for initial screening and randomization. Data collection was performed primarily by JG with MZ performing the data cleaning.

### 2.4. Data Characterization

Sample TikTok posts were categorized into one of nine “Nutrition Topics” that best reflected their primary nutrition focus and one of eight “Content Creator Type” descriptions that captured the creators’ overall content focus as well as their expertise or scope of practice (see Appendix A descriptions of “Nutrition Topics” and “Content Creator Type”). These categories were informed by a study by Denniss, Lindberg [25] but were then adjusted to better capture TikTok’s content landscape after the researchers familiarized themselves with TikTok’s unique environment. For example, “Content farms” was introduced as a “Content Creator Type” to describe the proliferation of accounts that produce low-quality content that exploit algorithms to maximize engagement [5]. To ensure the accuracy and reliability of categorization, posts were independently reviewed by three researchers based on the defined criteria, with discrepancies between findings being discussed in order to achieve consensus.

### 2.5. Evaluation for Quality

Sample TikTok posts were evaluated using a modified version of the Social Media Evaluation Checklist, originally developed by Squires, Brighton [26] to audit Australian dietitians’ social media activity for ethical and professional standards. The checklist assesses five key domains: (1) authorship (whether the creator’s credentials are disclosed), (2) accuracy (alignment with evidence-based dietary guidelines), (3) financial disclosure (disclosure of affiliations, sponsorships, or promotional intent), (4) transparency (clarity and reliability of shared information), and (5) engagement (the level of user interaction and potential for misinformation spread). Posts were scored as ‘met’, ‘not met’, ‘unsure’, or ‘not applicable’ to assess how closely content creators and their posts adhered to quality criteria. Modifications included removing dietitian-specific criteria and refining the accuracy assessment to fit the TikTok short-video format. These adjustments were made, as these criteria were found to be non-applicable when transferring the audit tool from a professional context to an account analyzing a broader range of content creators. This tool was selected, as it provided a framework to evaluate nutrition-related social media content, with quality criteria similar to previous health-related information evaluation tools [27], but in a more time-efficient manner.

### 2.6. Evaluation for Levels of Accuracy and Evidence

The accuracy of nutrition-related information in social media posts was evaluated using the research protocol developed by Denniss, Lindberg [25]. This involved evaluating content against evidence-based nutrition guidelines, including the Australian Dietary Guidelines, Nutrient Reference Values, and Practice-based Evidence in Nutrition. As per the protocol, the accuracy of the information presented was scored as “completely inaccurate”, “mostly inaccurate, some accuracies”, “mostly accurate, some inaccuracies”, “completely accurate” and “not assessable” (Appendix A) [25]. The level of evidence available supporting the nutrition information presented was graded on a scale from A–D, with “Grade A” indicating a strong body of evidence and “Grade D” indicating a weak body of evidence using the criteria for analyzing evidence outlined by the National Health and Medical Research Council (Appendix A) [28]. For these criteria, posts could also be categorized as “N/A” for “not assessable” by the protocol due to nutrition-related content describing a personal experience rather than relaying nutrition information or advice and, therefore, being unsuitable to assess for evidence and accuracy.

### 2.7. Statistical Analysis

Statistical analysis was conducted using Microsoft Excel. Descriptive statistics were used to summarize the background information of content creators posting nutrition-related content, the common nutrition-related themes and topics, and engagement metrics. Descriptive statistics were also used to capture the quality of the 250 sample TikTok posts, as determined by the Social Media Evaluation Checklist, as well as their level of accuracy and level of evidence in relaying nutrition information by nutrition topics and evidence-based information. Descriptive statistics (mean, standard deviation, range) were calculated for engagement metrics across accuracy categories. Independent *t*-tests were used to compare engagement between accurate and inaccurate posts, while ANOVA assessed differences in engagement across content creator types and nutrition topics. Chi-square tests were performed to examine associations between content accuracy and creator types. The statistical significance level was set at *p* < 0.05 for all tests.

The study adhered to ethical guidelines for research involving publicly available data. User identities were anonymized to protect privacy, and no personal identifying information was recorded.

## 3. Results

Of the 250 posts analyzed for this study, the most common content creator types publishing nutrition-related content on TikTok were health and wellness influencers (32%), followed by fitness content creators (18%) and “other” creators (18%) whose content does not typically focus on nutrition or health such as lifestyle content creators. By contrast, dietitians (5%), nutritionists (4%), and other health professionals (3%) were less frequently represented. Regarding the content of TikTok posts, the most prevalent nutrition topics featured were weight loss content (34%), recipes, meal ideas, and “What I Eat in a Day” videos (32%), followed by general nutrition advice (11%), foods, nutrients, and supplements (10%), and goal-oriented nutrition (7%).

### 3.1. Quality of Nutrition-Related TikTok Posts

When evaluating the quality of the TikTok posts using the criteria outlined by the Social Media Evaluation Checklist, 48% of applicable the TikTok posts did not meet the criteria for not using testimonials (Figure 2). The majority of the applicable TikTok posts did not meet the criteria requiring posts to include transparent advertising (82%) or to identify potential conflicts of interest (77%). Most of the posts on TikTok also failed to meet the criteria for not promoting stereotypical attitudes (63%), while around half of the applicable posts on TikTok also failed to meet the criteria for providing evidence-based information (55%). Nearly all the TikTok posts were found to have accurately reflected their message (98%). Additionally, most of the posts on TikTok (75%) failed to provide balanced accurate information or outline the risks and benefits associated with products and behaviors (90%).

### 3.2. Accuracy and Evidence Grade of Nutrition-Related TikTok Posts

When analyzing the 250 TikTok posts for accuracy, 14% were classified as completely accurate, with 29% of the posts containing mostly accurate information, 18% containing mostly inaccurate information, and 19% deemed completely inaccurate (Figure 3a and Figure 4a). A further 20% of the posts were considered not assessable under the accuracy criteria due to presenting nutrition content with no information, advice, or commentary. Regarding the level of evidence of the TikTok posts, 12% contained Grade A evidence (strongest) and another 12% used Grade B evidence (Figure 3b and Figure 4b). However, a larger proportion were assessed as relying on weaker evidence, with 20% featuring Grade C evidence and 15% Grade D evidence (weakest). Notably, 41% of posts were considered not assessable by the evidence criteria due to presenting content with an anecdotal or experiential slant.

When analyzing the quality of TikTok posts by nutrition topic (Figure 3a), posts discussing general nutrition information had the highest proportion of accurate and mostly accurate content (29% and 43%, respectively). Comparatively, posts focused on weight loss had the highest proportion of mostly inaccurate and completely inaccurate content (18% and 29%, respectively). Food, nutrient, and supplement posts were predominantly a mix of mostly accurate (46%) and mostly inaccurate (33%) information as were the accuracy levels of the goal-oriented nutrition content (41% and 29%, respectively). For the recipes and meal ideas, 50% of the posts were considered not assessable as they did present nutrition advice or information.

As seen in Figure 3b, general nutrition advice was primarily classified as containing Grade A and B evidence (32% and 25%, respectively). Food, nutrient, and supplement content predominantly fell under Grade C (38%) and Grade B (25%). The level of evidence for goal-oriented nutrition information or advice was primarily graded as C (53%), with a smaller proportion classified as A (24%). The higher proportion of high protein and weight loss posts that were considered non-assessable under the evidence criteria (44% and 29%, respectively) can be attributed to the inclusion of recipes, anecdotal content, or personal experiences rather than evidence-based information or nutrition advice.

Amongst content creators, dietitians were generally found to produce the highest proportion of completely accurate and mostly accurate posts (42% and 25%, respectively) (Figure 4a). By contrast, “content farms,” had the highest proportion of mostly inaccurate and inaccurate content (18% and 30%, respectively). Health and wellness creators showed 35% mostly accurate content but had 25% completely inaccurate content, while fitness creators had 32% mostly accurate, 23% mostly inaccurate, and 18% inaccurate content. Food content creators, nutritionists, and “other” creators had higher proportions of content considered not assessable.

When grading the level of evidence provided (Figure 4b), dietitians exhibited the highest proportion of Grade A content (50%), while “content farms” primarily consisted of Grade D evidence (30%). Health and wellness creators similarly demonstrated a mix of Grade C and D evidence levels (29% and 20%, respectively).

### 3.3. Engagement Metrics

Engagement metrics were analyzed as average number of likes, comments, saves, and shares by level of accuracy and evidence for assessable TikTok posts (Table 1). When evaluated against level of accuracy, mostly accurate posts (*n* = 72) had the highest average engagement metrics with an average of 108,391 likes (±339,711), 444 comments (±981), 6444 shares (±41,677), and 29,322 saves (±131,744). Completely accurate posts (*n* = 33) received 71,589 likes (±170,493), 240 comments (±392), 3444 shares (±7348), and 31,051 saves (±74,003). Completely inaccurate posts (*n* = 47) had 57,191 likes (±122,798), 476 comments (±1892), 6511 shares (±23,880), and 22,217 saves (±50,572), with mostly inaccurate posts having the lowest engagement with 55,630 likes (±105,342), 513 comments (±1158), 5268 shares (±16,864), and 17,260 saves (±40,329).

When evaluated against levels of evidence, Grade A posts (*n* = 31) had the highest engagement, with an average of 117,470 likes (±421,054), 345 comments (± 856), 13,107 shares (±63,439), and 48,868 saves (±200,901), while Grade C posts (*n* = 49) had the lowest engagement, with 40,168 likes (±78,268), 434 comments (±1100), 4061 shares (±15,867), and 13,846 saves (±36,384).

No statistical significance was observed in engagement metrics across different levels of accuracy or evidence (*p* > 0.05) (Appendix A).

## 4. Discussion

This study highlights the prevalence of nutrition-related content on TikTok, emphasizing the platform’s role as a significant source of health information for adolescents. TikTok is dominated by non-expert content creators, with health and wellness influencers comprising 32% of the sample. These creators, while popular, often disseminate information that lacks scientific credibility. In contrast, dietitians, who represented only 5% of the analyzed posts, produced the most accurate content (42% completely accurate). This disparity underscores the need to amplify the voices of qualified professionals on the platform. The prevalence of weight-normative and diet culture content, particularly posts focusing on weight loss (34%), raises concerns about the platform’s impact on adolescent and young adult health. Weight loss posts had the highest proportion of completely inaccurate information (28%), often promoting unhealthy or unsustainable practices. These findings align with previous research highlighting the risks of misinformation on body image and mental health among young users [29].

Engagement metrics revealed a troubling trend: completely inaccurate posts garnered significantly higher likes, comments, shares, and saves than accurate posts (*p* < 0.05). This suggests that TikTok’s algorithm amplifies sensationalized and engaging content, regardless of its accuracy. The platform’s algorithmic bias, combined with users’ preference for visually appealing or relatable content, may explain the popularity of inaccurate posts. Sensationalized or controversial nutrition claims often attract more attention, particularly on a platform driven by virality.

The results corroborate prior studies on the prevalence of misinformation on social media, particularly those identifying Instagram and YouTube as major sources of dietary misinformation [30,31,32]. However, TikTok’s unique algorithm and short-form video format appear to exacerbate the issue by rewarding virality over credibility. Like on other platforms, when using the TikTok platform, users are more likely to engage with entertaining or sensational content, even if it lacks accuracy [33,34].

The dominance of non-expert creators and the high engagement with inaccurate posts highlight the risks TikTok poses to adolescent and young adult health [4,21]. Research suggests that while official health institutions such as the Centers for Disease Control and Prevention (CDC), the American Dietetic Association (ADA), and the World Health Organization (WHO) provide highly accurate, evidence-based dietary recommendations, their content often lacks the engagement-driven appeal of social media influencers and viral posts [34]. A study comparing engagement across platforms found that scientific organizations struggle to match the reach of influencer-driven content on TikTok, despite providing more credible information [35]. This discrepancy highlights the challenge of ensuring that evidence-based nutrition guidance competes with algorithm-driven misinformation. Future research could explore strategies for increasing the visibility and engagement of expert-driven content on platforms such as TikTok. Future research could explore whether content creators deliberately manipulate dietary messages for engagement purposes, leveraging unconventional claims to appeal to users. Exposure to harmful nutrition messages can lead to disordered eating behaviors, body dissatisfaction, and poor dietary habits [36,37,38]. Given TikTok’s popularity among adolescents, these findings highlight the urgent need for targeted interventions to mitigate the spread of misinformation [39].

Social media platforms need to implement algorithmic adjustments to prioritize evidence-based content and reduce the visibility of sensationalized misinformation [33]. There is a need for tools to verify the credentials of content creators sharing health information [10]. Dietitians and other qualified professionals need to actively engage on TikTok to counteract misinformation with credible, engaging content [10]. Health professionals need to be trained on creating effective short-form videos that align with TikTok’s content style. Policymakers need to develop guidelines for social media platforms to regulate nutrition-related content and protect vulnerable populations, particularly adolescents [39].

Social media platforms, including TikTok, have introduced content moderation policies and fact-checking initiatives to mitigate misinformation. However, these measures primarily target political and medical misinformation, with limited enforcement in the nutrition and dietary space [40]. Unlike public health organizations, independent fact-checkers often lack the specific expertise needed to evaluate complex nutritional claims, leading to inconsistent content moderation. Moreover, TikTok’s policies do not prevent viral engagement with misleading nutrition content before fact-checking occurs, meaning misinformation often gains traction before it is flagged or removed [4,33]. Future research could assess the effectiveness of existing policies in limiting the spread of misleading dietary advice and explore new regulatory frameworks that balance content accessibility with public health interests.

Researchers can apply the findings to explore the mechanisms through which TikTok’s algorithm amplifies engagement with inaccurate nutrition content [5,33]. Investigating the role of visual appeal, sensationalism, and relatability in user interaction can provide insights into why misinformation thrives [41]. The results can inform future studies evaluating the short- and long-term effects of exposure to nutrition misinformation on adolescent and young adult dietary behaviors, mental health, and body image [36,42]. Longitudinal research can build on this study to identify trends and outcomes over time. Researchers can design and evaluate interventions aimed at improving media literacy among adolescents [43,44]. Studies assessing the impact of algorithmic adjustments, content moderation, or educational campaigns on reducing misinformation exposure are particularly relevant [18,20,45,46]. Future research can compare misinformation prevalence, engagement trends, and algorithmic influences across multiple platforms to identify platform-specific challenges and best practices.

Health professionals can use this study’s findings to develop engaging and accurate nutrition content tailored to TikTok’s format and audience preferences [9,47]. Posts promoting weight loss, while less accurate, demonstrated significant engagement due to their alignment with TikTok’s algorithmic promotion of visually appealing and sensational content. Effective use of short-form videos, trending audio, and visuals can enhance the reach and impact of credible content [48]. Dietitians and health educators can incorporate social media literacy into their practice, empowering adolescents to critically evaluate nutrition information on TikTok and other platforms [39]. This approach encourages informed decision making and reduces susceptibility to misinformation. Health professionals can collaborate with influencers who have established credibility among young people to amplify evidence-based messages [39]. Partnerships with trusted creators can help bridge the gap between expert knowledge and audience engagement.

On the other hand, policymakers could advocate for clearer content moderation guidelines on TikTok, encouraging the platform to prioritize evidence-based posts in its algorithm and limit the visibility of misleading or harmful content [13,31,32]. Credential verification for content creators discussing nutrition and health topics is important. Policies encouraging platforms to implement verified badges or disclaimers for expert content can help users identify credible sources [27,49]. There is a need for initiatives that increase digital and health literacy among young people, focusing on identifying and addressing misinformation [43]. Partnerships between schools, governments, and social media platforms can deliver educational programs effectively. Advertising regulations are also warranted to limit harmful marketing targeting vulnerable populations.

This study’s strengths include its robust sample size and the use of evidence-based evaluation frameworks, which enhance the reliability of findings. This research was the first to examine general nutrition trends prevalent on TikTok, thus providing groundbreaking insight into the dynamic landscape of the platform. Identifying the types of misinformation can be used to further develop practical recommendations to reduce the harms of misinformation. However, several limitations warrant consideration. First, the study focused solely on public-facing posts, potentially excluding private or regional content. Second, the reliance on keywords may have overlooked posts using alternative hashtags or algospeak. A key limitation of this study is the influence of TikTok’s algorithm on search results. Despite using newly created accounts in a private browser to minimize personalization, the platform’s dynamic content recommendation system likely influenced the types of posts retrieved. This may have led to an overrepresentation of highly engaged, viral, or controversial posts, while more accurate but lower-engagement content may have been underrepresented. Future research could explore alternative sampling strategies, such as API-based data collection or systematic tracking of content over time in order to better understand the influence of algorithmic curation on social media nutrition misinformation.

Finally, the cross-sectional design provides a snapshot of TikTok’s content landscape but does not capture longitudinal trends. Although engagement metrics were analyzed, comments were not as rigorously scrutinized. Analyzing comments in future could further provide insight into how users engage with misinformation. Future research should also explore longitudinal patterns in nutrition misinformation to understand how trends evolve and whether platform policies effectively curb the spread of false claims over time. Additionally, investigating user behaviors and motivations related to engaging with nutrition content—such as whether users seek out expert opinions or gravitate toward viral but misleading information—could provide deeper insights into the broader implications of social media-driven health misinformation.

## 5. Conclusions

This study demonstrates that TikTok’s nutrition-related content is heavily influenced by non-expert creators and diet culture, with misinformation often receiving higher engagement than credible information. Addressing these challenges requires collaborative efforts among social media platforms, health professionals, and policymakers to prioritize accurate, evidence-based nutrition messaging. Protecting adolescent and young adult health in the digital age demands targeted strategies to mitigate misinformation and promote positive health behaviors.

## Figures and Tables

**Figure 1 nutrients-17-00781-f001:**
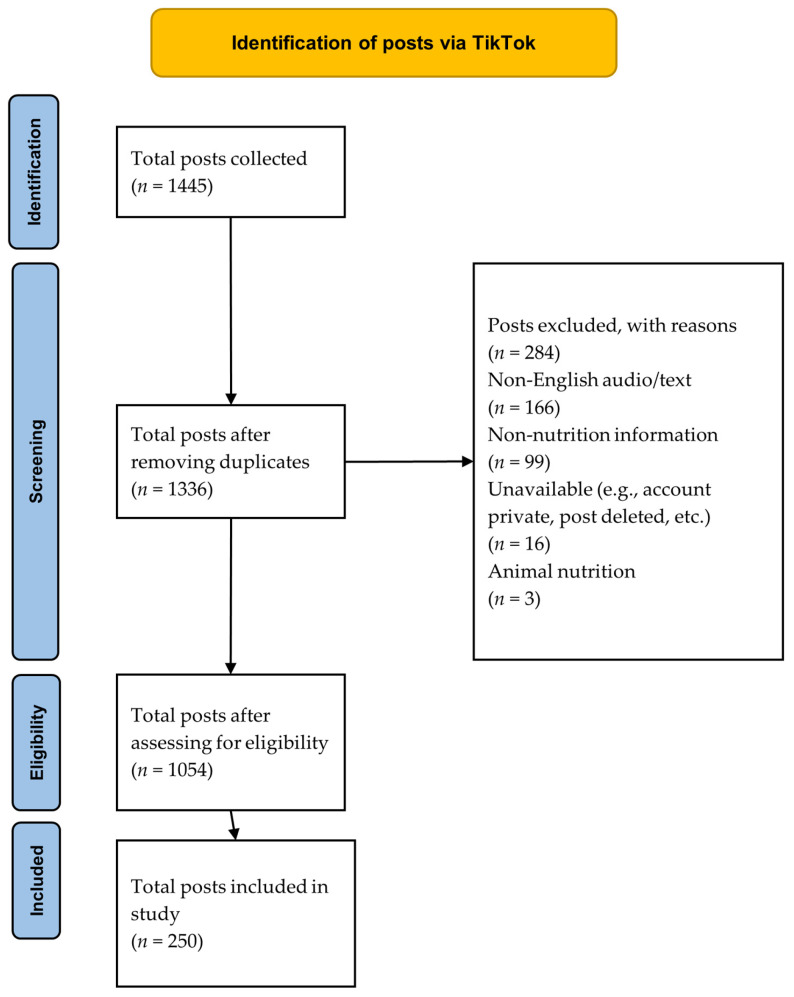
PRISMA flowchart illustrating screening of sample TikTok posts.

**Figure 2 nutrients-17-00781-f002:**
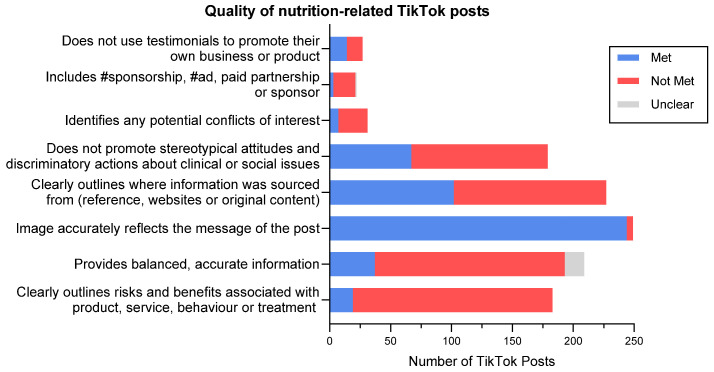
Quality of nutrition-related TikTok posts as defined by the Social Media Evaluation Checklist [26].

**Figure 3 nutrients-17-00781-f003:**
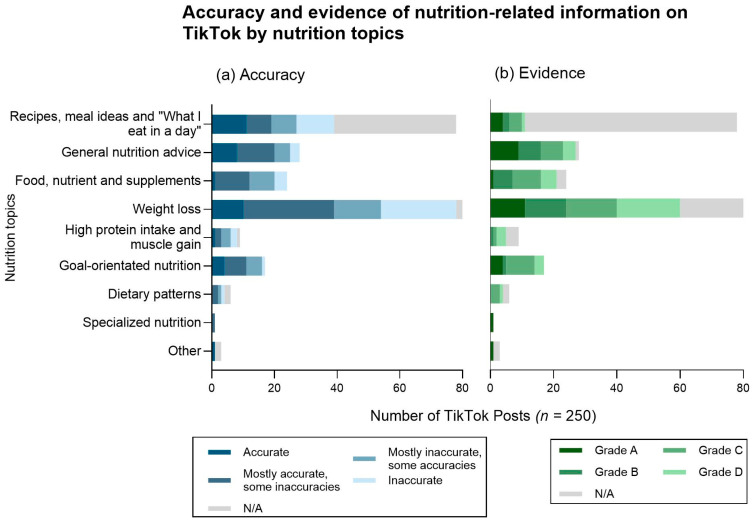
Distribution of nutrition topics in nutrition-related TikTok posts by levels of (**a**) accuracy and (**b**) evidence.

**Figure 4 nutrients-17-00781-f004:**
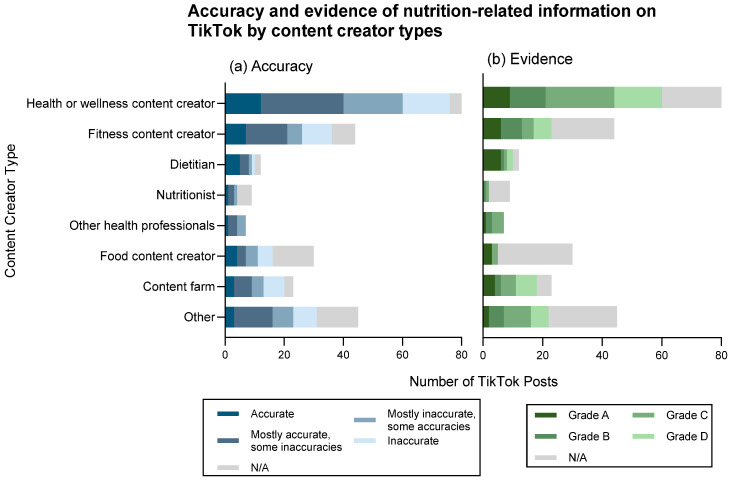
Distribution of content creator types publishing nutrition-related TikTok posts by levels of (**a**) accuracy and (**b**) evidence.

**Table 1 nutrients-17-00781-t001:** Average engagement metrics (likes, comments, saves, shares) for nutrition-related TikTok posts considered assessable by the categorization frameworks by levels of accuracy and evidence.

Levels of Accuracy and Supporting Evidence	Number of Assessable Posts, *n* (%)	Average Number of Likes, *n* ± SD	Average Number of Comments, *n* ± SD	Average Number of Saves, *n* ± SD	Average Number of Shares, *n* ± SD
Level of accuracy					
Total assessable posts, *n* = 250	197 (100)	77,967 ± 230,155	430 ± 1222	25,265 ± 90,351	5651 ± 28,749
Completely accurate	33 (17)	71,589 ± 170,493	240 ± 392	31,051 ± 74,003	3444 ± 7348
Mostly accurate, some inaccuracies	72 (37)	108,391 ± 339,711	444 ± 981	29,322 ± 131,744	6444 ± 41,677
Mostly inaccurate, some inaccuracies	45 (23)	55,630 ± 105,342	513 ± 1158	17,260 ± 40,329	5268 ± 16,864
Completely inaccurate	47 (24)	57,191 ± 122,798	476 ± 1892	22,217 ± 50,572	6511 ± 23,880
Level of evidence					
Total assessable posts, *n* = 250	147 (100)	86,251 ± 234,237	442 ± 1171	27,481 ± 88,551	6420 ± 28,825
Grade A	31 (21)	117,470 ± 421,054	345 ± 856	48,868 ± 200,901	13,107 ± 63,439
Grade B	30 (20)	95,404 ± 314,267	392 ± 966	15,670 ± 37,231	1096 ± 1942
Grade C	49 (33)	40,168 ± 78,268	434 ± 1100	13,846 ± 36,384	4061 ± 15,867
Grade D	37 (25)	70,386 ± 140,222	677 ± 2113	18,146 ± 48,422	7449 ± 26,692

## Data Availability

The raw data supporting the conclusions of this article will be made available by the authors on request.

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
