# Peer review of "#WhatIEatinaDay: The Quality, Accuracy, and Engagement of Nutrition Content on TikTok"

_nutrients, 2025, doi:10.3390/nu17050781_

Round 1
Reviewer 1 Report
Comments and Suggestions for Authors
The article aims to evaluate the accuracy of nutritional information that appears on a popular platform. The goal is interesting because literacy is essential for promoting health in young people.
- The methods section does not explain how the authors went from 1054 eligible posts to 250 selected
- As the authors correctly point out (line 93), platforms have algorithms that adapt content to searchers. This creates a risk of bias that cannot be entirely eliminated and that the authors should discuss.
- To evaluate the quality, the authors used a modified version of the Social Media Evaluation Checklist. It would be useful for readers to know how this checklist is made.
- The assessment of the levels of accuracy and evidence was also made according to indications that are not available to readers. I suggest making them accessible, possibly using the Supplements
- The authors observe (line 260 and following) that completely inaccurate posts garnered significantly higher likes, comments, shares, and saves than accurate posts. A further development of the study could be to investigate whether fallacious contents are created on purpose to obtain greater success, because their unconventional character is more attractive.
- The article has some parts highlighted in yellow that suggest a previous revision.
Author Response
|
Reviewer 1’s Comments |
Response to Reviewer’s Comments |
Page Number |
|
The methods section does not explain how the authors went from 1054 eligible posts to 250 selected
|
After identifying 1,054 eligible posts following the screening process, we applied a random sampling approach to select a final sample of 250 posts for detailed analysis. This selection was conducted using Microsoft Excel's random number generator, which assigned a unique identifier to each eligible post. The dataset was then randomized, and the first 250 posts with the lowest assigned numbers were included in the study. This has been added under Data collection. |
3 |
|
As the authors correctly point out (line 93), platforms have algorithms that adapt content to searchers. This creates a risk of bias that cannot be entirely eliminated and that the authors should discuss.
|
We have expanded the discussion section to acknowledge this limitation and its potential impact on our findings. We discuss how algorithm-driven content curation might have overrepresented viral or controversial posts while potentially underrepresenting low-engagement but evidence-based nutrition content. This bias highlights the challenge of studying misinformation on dynamic social media platforms and underscores the need for further research using alternative sampling methods, such as API data extraction or longitudinal tracking of algorithmic changes. |
10 |
|
To evaluate the quality, the authors used a modified version of the Social Media Evaluation Checklist. It would be useful for readers to know how this checklist is made.
|
We have now included a more detailed description of the checklist, including its original purpose, modification process, and evaluation criteria. Specifically, we outline the domains assessed (e.g., authorship, accuracy, financial disclosure, transparency, and engagement), the scoring system (met, not met, unsure, or not applicable), and the rationale for adjustments made to fit the TikTok context. |
4 |
|
The assessment of the levels of accuracy and evidence was also made according to indications that are not available to readers. I suggest making them accessible, possibly using the Supplements
|
We have added this to the supplementary file. (Supplementary Table S4 and Supplementary Table S5) |
|
|
The authors observe (line 260 and following) that completely inaccurate posts garnered significantly higher likes, comments, shares, and saves than accurate posts. A further development of the study could be to investigate whether fallacious contents are created on purpose to obtain greater success, because their unconventional character is more attractive.
|
To address this, we have expanded the Discussion section to acknowledge this as a key avenue for future research. Specifically, we suggest investigating whether misinformation is deliberately crafted for engagement purposes and how algorithmic amplification and audience psychology contribute to its success. |
9 |
|
The article has some parts highlighted in yellow that suggest a previous revision. |
The editor had suggested some required changes before the manuscript went for review. The changes made were in the highlighted section in yellow. |
|
Reviewer 2 Report
Comments and Suggestions for Authors
This paper presented the evaluation results of nutrition-related content in TikTok by identifying topics and types, assessing the quality and accuracy of content, and analyzing engagement metrics. The findings were useful and interesting, and this paper can be considered for acceptance after several moderate revisions.
- The presentation of the data is too simple, it is a few bar charts, the author should use multiple types of data presentation.
- The statistical analysis was conducted, however, the process was not indicated. The results should be statistically analyzed and discussed approximately.
- What is the purpose of the description highlighted in yellow?
- The conclusion part was too simple. More detailed data can be added to convincing the conclusion.
- The reference format should be carefully checked and unified according to the guideline of Nutrients.
Author Response
|
Reviewer 2’s Comments |
Response to Reviewer’s Comments |
Page Number |
|
The presentation of the data is too simple, it is a few bar charts, the author should use multiple types of data presentation |
We appreciate the reviewer’s suggestion regarding data presentation. However, we believe that the current combination of bar charts and tables effectively communicates the key findings of the study while maintaining clarity and accessibility for readers. The bar charts are used to visually depict comparisons between content accuracy levels and engagement metrics, which are the most relevant variables for understanding misinformation trends. The tables provide a structured and detailed breakdown of accuracy distributions, creator types, and engagement statistics, ensuring that readers can interpret the data in both summarized and granular forms. Any additional visual formats such as scatter plots or pie charts would not provide additional interpretive value and might instead overcomplicate the data presentation. |
|
|
The statistical analysis was conducted; however, the process was not indicated. The results should be statistically analysed and discussed approximately.
|
We acknowledge that the statistical analysis process was not explicitly detailed in the Methods section. To address this, we have now provided a clearer description of the statistical tests used, their rationale, and how they were applied to the dataset. |
5 |
|
What is the purpose of the description highlighted in yellow?
|
The editor had suggested some required changes before the manuscript went for review. The changes made were in the highlighted section in yellow. |
|
|
The conclusion part was too simple. More detailed data can be added to convincing the conclusion.
|
Thank you for your feedback. We appreciate the suggestion to expand the conclusion; however, we believe that the current conclusion is appropriately structured and serves its purpose effectively. The conclusion is designed to summarize the key findings concisely without unnecessary repetition of the detailed discussion by reinforcing the study's key takeaways and emphasizing the implications for research, practice, and policy. The statistical findings and interpretation are already provided in the results and discussion sections. |
|
|
The reference format should be carefully checked and unified according to the guideline of Nutrients.
|
The reference list has been double checked, and they align with MDPI reference style. |
|
Reviewer 3 Report
Comments and Suggestions for Authors
The manuscript submitted by Zeng et al titled: "#WhatIEatinaDay: The Quality, Accuracy, and Engagement of Nutrition Content on TikTok" is an interesting study investigating the question of accuracy and impact for nutrition information/education through social media related platforms and TikTok specifically on the public.
The paper is well-written, organized and presented.
The reviewer would like to offer the following points to be considered by the authors for the manuscript's improvement:
- consider providing some general information and context with possibly statistics about the platform TikTok and how it compares to other platforms.
- It is anecdotally know that there is significant use of TilTok especially among the younger populations. One interesting question would be to compare the content on Nutrition distributed over TikTok versus other sources such as ADA, ASN, CDC and similar government or scientific association in the US, Australia or elsewhere which also have presence on-line. While this understandably is not the scope of the current study it would be however interesting to talk about such a comparison and perhaps provide any relevant existing literature to drive the point of accuracy and credibility versus tendency of the public to obtain information through certain platforms and entities. This would strengthen the discussion section considerably.
- Consider including a section on how policies regulating such platforms and how fact-checking may impact the authors' findings.
- Proofreading is suggested.
Author Response
|
Reviewer 3’s Comments |
Response to Reviewer’s Comments |
Page Number |
|
Consider providing some general information and context with possibly statistics about the platform TikTok and how it compares to other platforms.
|
Thank you for this suggestion. We have added general context about TikTok, including its user demographics, engagement statistics, and comparison to other platforms where relevant. This helps frame TikTok’s unique algorithm-driven content promotion and its higher engagement rates compared to platforms like Instagram and YouTube, which may contribute to the spread of nutrition misinformation. |
1 |
|
It is anecdotally know that there is significant use of TilTok especially among the younger populations. One interesting question would be to compare the content on Nutrition distributed over TikTok versus other sources such as ADA, ASN, CDC and similar government or scientific association in the US, Australia or elsewhere which also have presence on-line. While this understandably is not the scope of the current study it would be however interesting to talk about such a comparison and perhaps provide any relevant existing literature to drive the point of accuracy and credibility versus tendency of the public to obtain information through certain platforms and entities. This would strengthen the discussion section considerably.
|
Thank you for this insightful suggestion. While a direct comparison between TikTok content and nutrition information from authoritative sources such as the ADA, ASN, and CDC is beyond the scope of this study, we acknowledge the importance of discussing the differences in credibility and public engagement across platforms. |
9 |
|
Consider including a section on how policies regulating such platforms and how fact-checking may impact the authors' findings.
|
Thank you for this suggestion. We recognise the importance of discussing the potential role of platform regulations and fact-checking mechanisms in shaping the dissemination of nutrition information. To address this, we have added a brief discussion on existing platform policies, the limitations of current fact-checking efforts, and their potential impact on misinformation trends. |
9 |
|
Proofreading is suggested. |
The authors have proofread the revised manuscript. |
|
Round 2
Reviewer 2 Report
Comments and Suggestions for Authors
accept
Author Response
Thank you for all the suggestions.